# Acute kidney injury in the UK: a replication cohort study of the variation across three regional populations

Simon Sawhney,[1,2] Heather A Robinson,[2,3] Sabine N van der Veer,[2,3] Hilda O Hounkpatin,[2,4] Timothy M Scale,[2,5] James A Chess,[2,5] Niels Peek,[2,3] Angharad Marks,[1,2] Gareth Ivor Davies,[2,5] Paolo Fraccaro,[2,3] Matthew J Johnson,[2,4] Ronan A Lyons,[2,5] Dorothea Nitsch,[2,6] Paul J Roderick,[2,7] Nynke Halbesma,[2,8] Eve Miller-Hodges,[2,9] Corrinda Black,[1,2] Simon Fraser[2,7]

## ABSTRACT

**Objectives** A rapid growth in the reported rates of acute kidney injury (AKI) has led to calls for greater attention and greater resources for improving care. However, the reported incidence of AKI also varies more than tenfold between previous studies. Some of this variation is likely to stem from methodological heterogeneity. This study explores the extent of cross-population variation in AKI incidence after minimising heterogeneity.

**Design** Population-based cohort study analysing data from electronic health records from three regions in the UK through shared analysis code and harmonised methodology.

**Setting** Three populations from Scotland, Wales and England covering three time periods: Grampian 2003, 2007 and 2012; Swansea 2007; and Salford 2012.

**Participants** All residents in each region, aged 15 years or older.

**Main outcome measures** Population incidence of AKI and AKI phenotype (severity, recovery, recurrence). Determined using shared biochemistry-based AKI episode code and standardised by age and sex.

**Results** Respectively, crude AKI rates (per 10 000/year) were 131, 138, 139, 151 and 124 (p=0.095), and after standardisation for age and sex: 147, 151, 146, 146 and 142 (p=0.257) for Grampian 2003, 2007 and 2012; Swansea 2007; and Salford 2012. The pattern of variation in crude rates was robust to any modifications of the AKI definition. Across all populations and time periods, AKI rates increased substantially with age from ~20 to ~550 per 10 000/year among those aged <40 and ≥70 years.

**Conclusion** When harmonised methods are used and age and sex differences are accounted for, a similar high burden of AKI is consistently observed across different populations and time periods (~150 per 10 000/year). There are particularly high rates of AKI among older people. Policy-makers should be careful not draw simplistic assumptions about variation in AKI rates based on comparisons that are not rigorous in methodological terms.

## Strengths and limitations of this study

► Previous studies have reported substantial variation in the incidence of acute kidney injury (AKI) between regions and over time, but have involved heterogeneous methods that limit comparability. To our best knowledge, this is the first cross-population study of AKI incidence within one study, with minimised methodological heterogeneity by sharing analysis code across regions.

► By using consistent methods, and real-life, routinely collected healthcare data, we provide new evidence that the rates of AKI in the UK are similar across different regions and time periods: ~150 events per 10 000/year (1.5% of the population).

► These findings may not be generalisable outside of the regions of the UK in the study. However, to enable researchers to replicate this work, we have made publicly available our analysis code for identifying and characterising AKI episodes.

For numbered affiliations see end of article.

**Correspondence to**
Dr Simon Sawhney;
simon.sawhney@abdn.ac.uk

## INTRODUCTION

The reported outcomes following acute kidney injury (AKI) are consistently poor.[1]

Reports of a growth in rates of AKI have led to calls for greater attention and resources for improving care,[2] but there is a more than tenfold variation between studies in the reported population incidence of AKI.[3–7] Population-based estimates of AKI incidence range from 18 per 10 000/year[3] to 250 per 10 000/year[5] based on changes in serum creatinine over time, and from 3 to 40 per 10 000/year based on hospital episode codes for 'non-dialysis requiring AKI'.[8 9] This wide variation is difficult to fully explain,[10] but is likely to be due in part to a changing clinical landscape with evolving international AKI criteria,[11–14] and different pragmatic interpretations of AKI criteria in research.[5 15] These reasons for variation are all potential sources of bias in clinical studies of AKI (figure 1). Without a clearer understanding of why populations differ, it is challenging (and potentially misleading) to interpret clinical

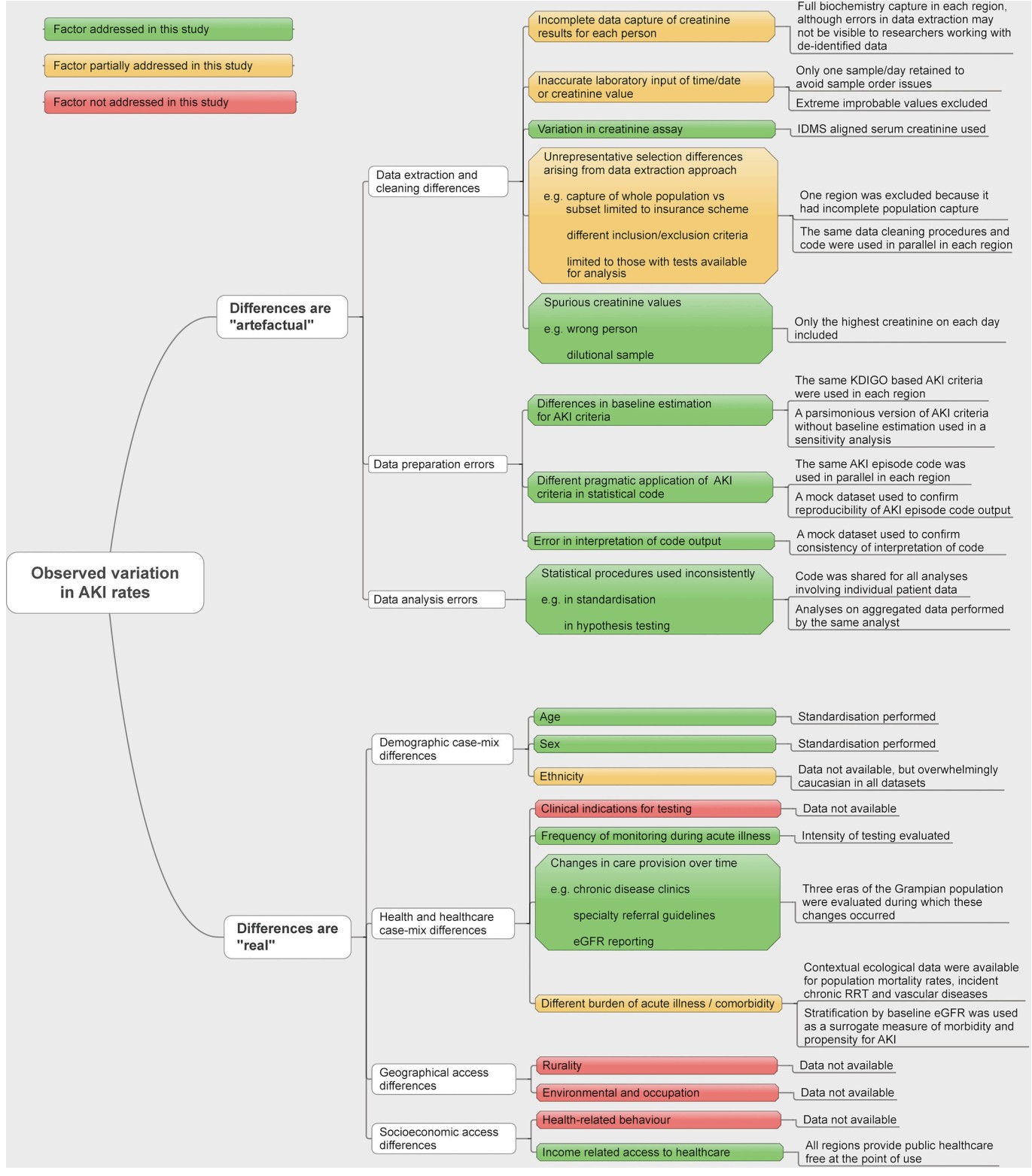

**Figure 1** Conceptual framework for the reasons for cross-population differences in acute kidney injury (AKI) rates. eGFR, estimated glomerular filtration rate; IDMS, isotope dilution mass spectrometry; KDIGO, Kidney Disease: Improving Global Outcomes; RRT, renal replacement therapy.

research in context, to make comparisons across populations or over time, or to make informed public health recommendations.

Worldwide, health services are undertaking quality initiatives to increase clinical awareness and improve treatment of AKI[16–19] in order to achieve the International Society of Nephrology target of eliminating avoidable deaths from AKI by 2025.[20] To evaluate the effectiveness of these initiatives, it is vital that there is a harmonisation of approaches to clinical research. This

means minimising methodological heterogeneity so that the findings of future research are more comparable, and maximising transparency so that trends in disease incidence and outcomes can be understood. Methodological heterogeneity can arise when researchers extract data from different data infrastructures, make different assumptions and adopt different criteria for identifying events. These steps are particularly important in AKI because of the recognised challenges of AKI research: it occurs unpredictably, in different clinical locations,[21] may be transient[22] and relies on trends rather than absolute values.[12–14] Small differences in how these challenges are handled can alter both the reported incidence and prognosis of AKI.[5 15 21 23] Despite its importance, this information is often undocumented or described in insufficient detail for research to be reproduced.[24] We have described these reasons for variation in AKI rates in a conceptual model (figure 1).

Algorithms using blood test data from electronic health records (EHRs) offer the potential of an objective common language for observing common diseases in clinical practice, audit and research.[25] In previous work, we developed an extended version of a widely used National Health Service (NHS) algorithm for detecting AKI in blood tests,[26] which flags individual 'AKI' blood tests and also applies phenotyping methods to combine AKI-flagged blood tests into clinically meaningful AKI illness episodes grouped by severity, duration, recovery and recurrence.[27 28] Sharing this algorithm between researchers working with different populations provides an opportunity to develop a harmonised approach to clinical research, robustly comparing the burden of AKI across different populations and over time, even when patient-level data cannot be shared. We used this to study the variation in the incidence of AKI across three populations from England, Scotland and Wales. The analysis spans a decade of change in the clinical awareness of AKI,[16] change in international AKI criteria[12–14] and change in the emphasis on community surveillance of people with chronic diseases.[29–31] Our aim was to explore the extent of cross-population variation in AKI incidence using real-life data, while minimising heterogeneity through harmonised methods.

## MATERIALS AND METHODS
### Population profiles
This study compares datasets created using linked EHR data from primary and secondary care for three UK regions with different 'index' years from 2003 to 2014: Grampian 2003, 2007 and 2012; Abertawe Bro Morgannwg University Health Board (ABMU, referred to in this article as Swansea) 2007; and Salford 2012 (online supplementary figure 1). Each dataset involves health data from the UK NHS and includes complete primary and secondary care biochemistry capture for the region. A fourth region initially considered for this analysis (from South England) was excluded because initial inspection of the

data characteristics revealed that the population capture of the data source was incomplete and might have led to bias in the estimation of AKI. All regions provide public healthcare, free at the point of use.

NHS Grampian, a health authority in Scotland, is served primarily by one large tertiary hospital and another district general hospital. All biochemistry for the dataset was extracted from a single biochemistry department covering the entire regional population.[5] The Grampian dataset was linked with the Scottish Renal Registry to exclude those already receiving chronic renal replacement therapy (RRT), to avoid misclassification of RRT as AKI. Similarly, Salford (North England) represents one borough of Greater Manchester, served by a single NHS hospital and biochemistry laboratory.[32] Read codes (version 2) were used to extract biochemistry information and exclude records from people receiving chronic RRT. In contrast, ABMU (Swansea, Wales) in 2007 covered a region served by four district general hospitals and four laboratories using two information management systems.[33 34] Those receiving chronic RRT could not be directly determined from a register but could be excluded based on the hospital location marked on the blood tests.

To provide further contextual description of these populations, we collected information on population mortality and relevant morbidities (renal and vascular) from the Office of National Statistics, UK Renal Registry, and Quality Outcomes Framework (QOF) data entered by general practitioner (GP) practices (table 1). Importantly, QOF data represent incentivised recording by GPs of people with a given condition (eg, chronic kidney disease), rather than actual population prevalences. This means that small differences in prevalence on the disease registers may represent recording practice as well as actual disease prevalence and should be interpreted with caution.

### Conceptual framework
In figure 1, we provide a conceptual framework for understanding the sources of variation in AKI revealed by our analysis. We sought to minimise 'artefactual' methodological differences in AKI episode rates by using only datasets where complete data capture (from both hospital and community settings) was possible, by harmonising data preparation and cleaning, and by standardising code sets for identifying AKI episodes. We also accounted for 'real' potential sources of variation in AKI rates by performing age and sex standardisation, stratification by baseline eGFR for case-mix differences and comparing the number of people with blood tests in rapid succession as a surrogate for presence of an acute illness.

### Data extraction and processing
This study used a distributed analysis approach to protect the confidentiality of patient-level data. Data were analysed by on-site researchers working from the same code. Non-disclosive summary statistics were aggregated into a single dataset, which was analysed

**Table 1** Contextual information on the populations in this study

| | Grampian 2003 | Grampian 2007 | Grampian 2012 | Swansea 2007 | Salford 2012 |
|---|---|---|---|---|---|
| Midyear regional population (all ages) during index year* | 529 360 | 548 290 | 573 400 | 499 400 | 237 085 |
| Midyear regional adult population (age ≥15 years) during index year* | 438 332 | 458 900 | 482 444 | 415 500 | 193 882 |
| Percentage of population in urban settlements of >10 000 people† | 49.3% | 51.8% | 52.1% | 81.7% | 99.9% |
| Regional crude all-cause mortality rate ages 15+ (index year/100 000)* | 1192 | 1154 | 1093 | 1334 | 1135 |
| Crude adult incidence of chronic RRT per million population (UKRR)‡ | 98 | 102 | 93 | 167 | 85 |
| Prevalence of chronic kidney disease per 100 people (QOF)§ | NA¶ | 2.6 | 3.3 | 1.8 | 3.0 |
| Prevalence of coronary heart disease per 100 people (QOF)§ | 4.1 | 4.0 | 3.9 | 4.1 | 3.9 |
| Prevalence of diabetes registration per 100 people (QOF)§ | 3.0 | 3.3 | 4.2 | 4.3 | 4.5 |
| Prevalence of heart failure registration per 100 people (QOF)§ | NA§ | 0.8 | 0.8 | 1.0 | 0.9 |
| Prevalence of hypertension registration per 100 people (QOF)§ | 11.0 | 11.7 | 13.2 | 12.5 | 13.8 |
| Prevalence of stroke and TIA registration per 100 people (QOF)§ | 1.6 | 1.7 | 1.9 | 2.1 | 1.8 |
| No of biochemistry departments for whole region | One department covers in and outpatient, community and private tests | One department covers in and outpatient, community and private tests | One department covers in and outpatient, community and private tests | Four departments cover in and outpatients, community and private tests | One department covers in and outpatient and community tests. Privately obtained samples unavailable |
| Means of excluding samples belonging to people on long term RRT from dataset | Link to Scottish Renal Registry | Link to Scottish Renal Registry | Link to Scottish Renal Registry | Removing samples from locations where renal replacement is performed, including intensive care unit | Read code screening |
| IDMS aligned creatinine assay | Yes | Yes | Yes | From 2007 | Yes |

*From the Office of National Statistics.
†From the 2011 national census in England and Wales and Scottish government urban rural classification.
‡From the UK renal registry (UKRR) annual reports.
§Quality outcomes framework (QOF) data are incentivised information entered by general practitioner practices. Not recorded in Grampian in 2003, for which 2004 data are provided where available.
¶Data not available.
IDMS, isotope dilution mass spectrometry; NA, not applicable; RRT, renal replacement therapy; TIA, Transient ischaemic attack; UKRR, UK Renal Registry.

centrally. This ensured that patient-level data were never brought together in a single physical location. All serum creatinine results for each individual were extracted. Creatinine values that were missing, were a non-value (eg, 'sample inadequate', 'sample error') or were lower than the limit for detection of the analyser were excluded. The 'Modification of Diet in Renal Disease' study estimated glomerular filtration rate (eGFR) was calculated using the abbreviated four-variable equation.[35] Finally, to avoid a non-chronological evaluation of samples from different locations, where multiple samples were available for the same individual on a given day, the sample with the highest creatinine value was retained for analysis.

## AKI identification and phenotyping

A challenge of AKI clinical research is the operationalisation of precise international AKI criteria in 'real-life' data where people do not receive blood tests in a protocolised fashion. Blood tests may not have been done at the necessary times to directly observe an acute rise in creatinine from a previous baseline, and assumptions based on available data are required. We identified differences in assumptions for determining AKI as an important potential methodological reason for observed variation in AKI rates (figure 1) and therefore used the exact same definition and analysis code in each region. Kidney Disease: Improving Global Outcomes (KDIGO)-based AKI detection and phenotyping algorithm code was applied by separate analysts working locally on each dataset.[14] As summarised in online supplementary table 1, these criteria compare each blood test with previous 'baseline' results within the last 365 days ('the look-back period') to determine if a recent change has occurred.[27] Where AKI occurred, a 'look-forward period' of 90 days was used to follow and phenotype the whole AKI episode. In online supplementary figure 2, these look-back and look-forward time periods are illustrated for a single hypothetical patient with respect to a moment of developing AKI within the index year. For those without AKI, the first eGFR of the index year was used as the baseline eGFR. For convenience, we used a baseline eGFR <60 mL/min/1.72 m$^2$ as an indicator of chronic kidney disease. Shared Stata code provided the following outputs: number of blood tests consistent with AKI, number of AKI episodes, baseline eGFR, AKI episode severity stage, progression of AKI severity from a lower to higher stage, recovery to baseline within 90 days and presence of prior AKI episodes in the past 3 years (ie, making the episode a recurrent AKI episode).

We also analysed data using more parsimonious versions of the KDIGO criteria: a 'narrow interpretation' in which blood tests were only compared if they were no more than a week apart (ie, restricted to criteria 2 and 3), and a 'very narrow interpretation' comparing only tests no more than 2 days apart (ie, restricted to criterion 3). If variation was due to a lack of robustness of AKI criteria in the face of estimating baseline from less recent data, these narrower interpretations would be expected to lead to less variation in AKI incidence.

To ensure uniformity of the application and interpretation of AKI code, a mock dataset of 40 hypothetical patients was developed. This mock dataset deliberately contained unformatted variables and a variety of creatinine trend patterns to represent a full range of data cleaning steps, AKI phenotypes, blood test intervals and interpretation issues. Each analyst used the same code on the test dataset and reproduced the same results before progressing to analysing regional data. We have made the algorithm code, mock dataset and instructions for their optimal use in Stata freely available online (https://github.com/RenalHDRUK).

## Statistical analysis

Analyses included the description of baseline characteristics, comparison of both crude and age-sex standardised rates of AKI, and phenotypes of AKI episodes. We also compared AKI rates in subgroups of baseline eGFR (as described above) and individual components of AKI criteria to determine if variations in rate were robust to changes in the AKI definition (table 2). AKI can only be identified when sufficient blood tests have been performed to detect a change. Therefore, to evaluate reasons for residual variation, we described the patterns of blood testing in each region, including the frequency of blood tests, the regularity (eg, blood tests no more than 2 and 7 days apart) and blood test location (hospital and outpatient/community).

Baseline characteristics included age, sex, the number of people with evidence of renal impairment (eGFR <60 mL/min/1.73 m$^2$) on their first test in the index year and the number of people with blood tests sufficiently close together for it to be possible to detect an 'AKI' result if present (two tests no more than 365 days apart).

We compared population rates of AKI episodes across each region and index year. We compared AKI episode rates using national statistics midyear population estimates for each region and then standardised to the England population for 2012,[36] a reference population selected as two of the three regions provided 2012 data. All AKI episodes in the index year counted towards the overall AKI episode rate. One-way analysis of variance followed by Tukey's post hoc test (in the event of significant differences) was used to identify pairwise significant differences in population level AKI episode rates.

For people with sufficient blood tests to potentially detect an episode of AKI (at least two tests no more than 365 days apart), we compared rates within eGFR strata (<30, 30–44, 45–59 and ≥60 mL/min/1.73 m$^2$). The proportion of the population with at least one AKI result based on AKI criteria 1, 2 or 3 (table 2), and the proportion of the population with at least one AKI result based on narrower interpretations of KDIGO criteria (restricting to criteria 2 and 3, or criterion 3 alone) were also recorded. To evaluate the impact of incomplete biochemistry capture, we also recalculated AKI rates using only tests taken from people in hospital. Of note, a distinction between hospital inpatient and outpatient results was not possible in Salford.

To evaluate potential sources of residual variation in AKI rates after harmonised analysis, we compared patterns of blood testing (number, frequency and location).

## Patient involvement

No patients were involved in development of the research question or the design of the study. There are no plans to disseminate the results of the research to study participants.

**Table 2** Baseline characteristics for each dataset

| | Grampian 2003 | Grampian 2007 | Grampian 2012 | Swansea 2007 | Salford 2012 |
|---|---|---|---|---|---|
| | Patient total (%) | Patient total (%) | Patient total (%) | Patient total (%) | Patient total (%) |
| Adult resident population (aged ≥15) | 438 332 | 458 900 | 482 444 | 415 500 | 193 882 |
| **Population ascertainment of renal impairment (eGFR <60 mL/min/1.73 m$^2$) in index year** | | | | | |
| No tests during index year | 311 922 (71.2)* | 303 673 (66.2) | 301 992 (62.6) | 253 531 (61.0) | 116 977 (60.3) |
| eGFR ≥60† | 101 595 (23.2) | 120 854 (26.3) | 158 736 (32.9) | 129 959 (31.3) | 66 890 (34.5) |
| eGFR <60† | 24 805 (5.7) | 34 373 (11.3) | 21 716 (4.5) | 32 010 (7.7) | 10 015 (5.2) |
| **Sufficiency of tests to enable AKI detection** | | | | | |
| People with no tests during index year | 311 922 (71.2) | 303 673 (66.2) | 301 992 (62.6) | 253 531 (61.0) | 116 977 (60.3) |
| People with insufficient tests | 52 602 (12.0) | 57 788 (12.6) | 69 239 (14.4) | 59 839 (14.4) | 31 467 (16.2) |
| People with ≥2 tests within 365 days | 73 808 (16.8) | 97 439 (21.2) | 111 213 (23.1) | 102 130 (24.6) | 45 438 (23.4) |
| **Characteristics of people with ≥2 tests within 365 days** | | | | | |
| Proportion women | 40 413 (54.8) | 53 061 (54.5) | 60 330 (54.2) | 55 685 (54.5) | 24 723 (54.4) |
| Median age (IQR) | 63 (48–74) | 63 (50–75) | 63 (49–74) | 64 (51–75) | 63 (49–74) |
| eGFR <60† | 18 573 (25.2)‡ | 28 274 (29.0) | 18 679 (20.2) | 25 952 (25.4) | 8541 (18.8) |

*Expressed as a percentage of total residents unless specified otherwise.
†First eGFR in index year (mL/min/1.73 m$^2$).
‡Expressed as a percentage of people with ≥2 tests within 365 days.
AKI, acute kidney injury; eGFR, estimated glomerular filtration rate.

## RESULTS

### Populations and baseline characteristics

As described in table 1, populations ranged in size from 193 882 (Salford 2012) to 482 444 people (Grampian 2012) (table 3). Crude reported population mortality rates were higher in Swansea than Grampian and Salford, as was the incidence of people starting long-term RRT. The recognition of diabetes and cardiovascular diseases in incentivised GP registers was similar across the populations.

Table 2 shows the baseline characteristics of extracted datasets after harmonised data cleaning. The percentage of people with at least two tests no more than 365 days apart varied from 17% to 25% with the fewest in the earliest dataset (Grampian 2003). There was a greater proportion of people tested with renal impairment (eGFR <60 mL/min/1.73 m$^2$) in 2007 compared with the other years of study.

### Incidence of AKI episodes

Table 3 and figure 2 show the differences in crude and standardised rates of AKI episodes for each dataset. A minority of people had more than one AKI episode in the index year. For reporting AKI episode rates (table 3), all episodes are included, whereas for reporting phenotypes of people with an AKI episode, the first episode is described (bottom of table 3 and table 4). Crude AKI rates varied with the lowest in Salford 2012 and highest

rate in Swansea 2007 (124–151 per 10 000/year, p=0.095). Standardisation by age and sex accounted for residual differences (142–151 per 10 000/year, p=0.257), with 95% CIs overlapping in all instances. Age and sex standardised AKI rates varied little between Grampian 2003, 2007 and 2012 (146–151 per 10 000/year). Table 3 also shows that the majority of people developing AKI could be identified using hospital tests alone, and just over half could be identified in each region using a rigid interpretation of KDIGO AKI criteria. Finally, across all populations, the proportion of people developing AKI in the index year increased substantially with increasing age and lower eGFR.

As shown in figure 3, the pattern of variation in crude AKI rates was the same when narrower interpretations of KDIGO AKI criteria were used, comparing only blood tests in the prior 2 and 7 days. Table 4 shows this pattern was also similar when analysis was limited to each individual component of the AKI criteria, or within strata of baseline eGFR.

### AKI phenotypes

Table 4 describes the first AKI episode for people with an AKI episode during the index year. As well as having the highest crude AKI rate, a greater proportion of those with AKI in Swansea were older, had baseline eGFR <60 mL/min/1.73 m$^2$ (37.6%), had a severe AKI episode (15.4% stage 3) and had non-recovery at 90 days (45.1%). In

**Table 3** Crude and standardised rates of acute kidney injury (AKI) episodes, and components of AKI criteria

| | Grampian 2003 | Grampian 2007 | Grampian 2012 | Swansea 2007 | Salford 2012 |
| --- | --- | --- | --- | --- | --- |
| | (rate per 10 000)* | (rate per 10 000) | (rate per 10 000) | (rate per 10 000) | (rate per 10 000) |
| Adult resident population | 438 332 | 458 900 | 482 444 | 415 500 | 193 882 |
| **AKI incidence rates** | | | | | |
| Crude AKI incidence (95% CI) | 131.2 (127.7 to 134.7) | 138.3 (134.9 to 141.7) | 139.1 (135.8 to 142.4) | 151.1 (147.4 to 154.8) | 124.3 (118.8 to 129.8) |
| Age–sex standardised AKI incidence (95% CI) | 147.2 (143.3 to 151.1) | 150.6 (146.9 to 154.3) | 146.3 (142.8 to 149.8) | 145.6 (142.0 to 149.2) | 141.8 (136.2 to 147.4) |
| Total AKI episodes | 5749 (131) | 6346 (138) | 6711 (139) | 6266 (151) | 2399 (124) |
| People with AKI | 5362 (122) | 5930 (129) | 6277 (130) | 5847 (141) | 2208 (114) |
| **Subgroups of people with AKI** | | | | | |
| AKI using hospital tests only | 4386 (100) | 4739 (103) | 4492 (93) | 4432 (107) | NA† |
| Rigid KDIGO criteria | 3436 (78) | 3803 (83) | 3617 (75) | 3469 (83) | 1114 (57) |
| People meeting 2d criterion | 2486 (57) | 2831 (62) | 2714 (56) | 2424 (58) | 741 (38) |
| People meeting 7d criterion | 2488 (57) | 2698 (56) | 2664 (55) | 2611 (63) | 821 (42) |
| People meeting 8–90d criterion | 2619 (60) | 2830 (59) | 3351 (69) | 3287 (79) | 1163 (60) |
| People meeting 91–365d criterion | 1408 (32) | 1528 (32) | 1850 (38) | 1591 (38) | 737 (38) |
| **People with AKI in age strata** | | | | | |
| ≥70 years | 3205 (562) | 3561 (587) | 3705 (572) | 3785 (584) | 1299 (544) |
| 40–69 years | 1765 (88) | 1903 (89) | 2021 (89) | 1699 (89) | 740 (92) |
| <40 years | 392 (22) | 466 (25) | 551 (29) | 363 (23) | 169 (19) |
| **People with AKI in eGFR strata among people with at least two tests within 365 days (rates expressed within strata of tested individuals at risk)** | | | | | |
| Baseline eGFR ≥60 | 3612 (654) | 3874 (560) | 4419 (478) | 3648 (479) | 1512 (410) |
| Baseline eGFR 45–59 | 809 (673) | 940 (496) | 894 (756) | 1044 (618) | 323 (607) |
| Baseline eGFR 30–44 | 597 (1222) | 723 (1000) | 661 (1282) | 732 (1097) | 202 (867) |
| Baseline eGFR <30 | 344 (2064) | 393 (1861) | 303 (1781) | 423 (1778) | 171 (1921) |

*Rate expressed per 10 000 residents unless specified otherwise.
†Location data not available.
eGFR, estimated glomerular filtration rate; KDIGO, Kidney Disease: Improving Global Outcomes; NA, not applicable.

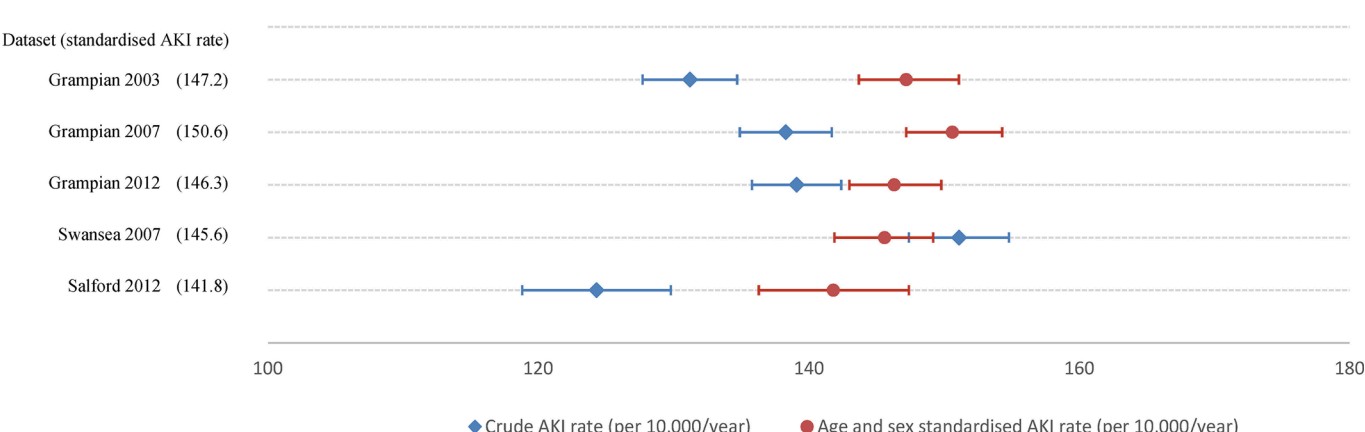

**Figure 2** Crude and age–sex standardised rate of acute kidney injury (AKI) episodes.

Grampian between 2003 and 2012, there was a steady improvement in the proportion of people with renal recovery 90 days after AKI from 42% to 49%.

**Further sources of variation**

In addition to assessing for age, sex and case-mix differences, we evaluated the blood testing patterns and

| Table 4 Phenotype of acute kidney injury (AKI) episodes | | | | | |
|---|---|---|---|---|---|
| | **Grampian 2003** | **Grampian 2007** | **Grampian 2012** | **Swansea 2007** | **Salford 2012** |
| | **Total people (%)** | **Total people (%)** | **Total people (%)** | **Total people (%)** | **Total people (%)** |
| People with AKI* | 5362 | 5930 | 6277 | 5847 | 2208 |
| Proportion women | 2899 (54.1) | 3256 (54.9) | 3443 (54.9) | 3195 (54.6) | 1250 (56.6) |
| Median age (IQR) | 73 (61–81) | 74 (61–82) | 74 (60–82) | 76 (64–84) | 74 (61–83) |
| **Peak AKI severity stage for first episode** | | | | | |
| Stage 1 | 3720 (69.4) | 4211 (71.0) | 4389 (69.9) | 3720 (63.6) | 1435 (65.0) |
| Stage 2 | 1014 (18.9) | 1063 (17.9) | 1174 (18.7) | 1224 (20.9) | 451 (20.4) |
| Stage 3 | 628 (11.7) | 656 (11.1) | 714 (11.4) | 903 (15.4) | 322 (14.6) |
| AKI stage progression | 817 (15.2) | 792 (13.4) | 850 (13.5) | 900 (15.4) | 300 (13.6) |
| **Baseline eGFR for first episode (mL/min/1.73 m$^2$)** | | | | | |
| ≥60 | 3612 (67.4) | 3874 (65.3) | 4419 (70.4) | 3648 (62.4) | 1512 (68.5) |
| 45–59 | 809 (15.1) | 940 (15.9) | 894 (14.2) | 1044 (17.9) | 323 (14.6) |
| 30–44 | 597 (11.1) | 723 (12.2) | 661 (10.5) | 732 (12.5) | 202 (9.1) |
| <30 | 344 (6.4) | 393 (6.6) | 303 (4.8) | 423 (7.2) | 171 (7.7) |
| **Prior AKI episodes detected in last 3 years** | | | | | |
| No prior episodes | 4415 (82.3) | 4847 (81.7) | 5052 (80.5) | 4824 (82.5) | 1708 (77.4) |
| One prior episode | 723 (13.5) | 833 (14.0) | 897 (14.3) | 784 (13.4) | 349 (15.8) |
| Two or more prior episodes | 224 (4.2) | 250 (4.2) | 328 (5.2) | 239 (4.1) | 151 (6.8) |
| Prior AKI within 1 year | 414 (7.7) | 459 (7.7) | 492 (7.8) | 488 (8.3) | 216 (9.8) |
| **Renal recovery to within 20% of baseline** | | | | | |
| Renal recovery | 2239 (41.8) | 2588 (43.6) | 3077 (49.0) | 2156 (36.9) | 970 (43.9) |
| Renal non-recovery | 2203 (41.1) | 2387 (40.3) | 2245 (35.8) | 2635 (45.1) | 820 (37.1) |
| Repeat samples not available | 920 (17.2) | 955 (16.1) | 955 (15.2) | 1056 (18.1) | 418 (18.9) |

*Numbers in brackets expressed as a percentage of people with at least one AKI episode.
eGFR, estimated glomerular filtration rate.

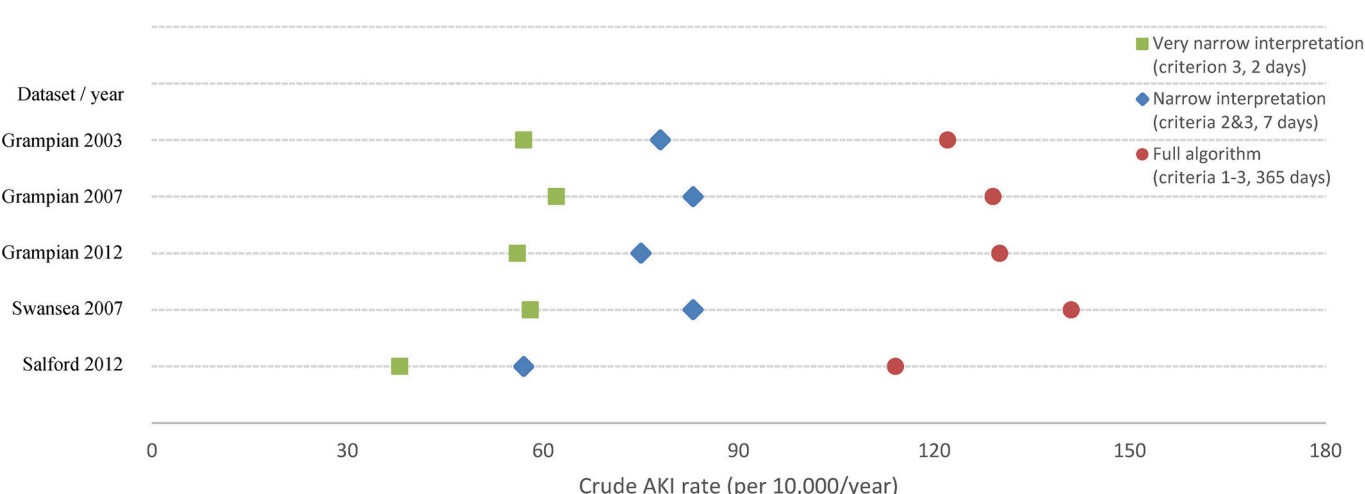

**Figure 3** Crude acute kidney injury (AKI) rates using different interpretations of the Kidney Disease: Improving Global Outcomes-based AKI definition.

clinical location contexts of each dataset (figure 4). Figure 4A shows the frequency of blood tests taken grouped by location: hospital inpatient or outpatient/community. Figure 4B shows the proportion of people with blood tests in close succession. In Grampian from 2003 to 2012, community blood testing increased over time, but the frequency of hospital inpatient testing remained unchanged. Test location was not available in Salford, but the proportions of people with two blood tests no more than 2 and 7 days apart was lower than in Grampian and Swansea. Figure 1 shows the conceptual framework for understanding these sources of variation.

**A**

Number of tests per resident

■ Inpatient tests  ■ Outpatient/community tests  ■ Location not available

**B**

Proportion of residents with ≥2 tests within the time frame

■ ≥2 tests within 7 days  ■ ≥2 tests within 2 days

**Figure 4** Patterns of blood testing by clinical location (A) and by test regularity (B).

## DISCUSSION

To our knowledge, this is the first multicentre study to systematically evaluate the extent of and reasons for regional and temporal variation in population rates of AKI, using a harmonised methodological approach. There were differences in the crude rates of AKI between datasets, but after accounting for age and sex, standardised rates were strikingly similar (at 140–150 episodes per 10 000/year, or ~1.5% of the population). The consistently high proportion of people aged over 70 developing AKI was also striking (>5%) and has implications for the planning the future healthcare requirements of an ageing population. This analysis shows the importance of both harmonised methods and standardisation for case-mix prior to any between-centre comparisons for description of variation in AKI.

Our analysis provides additional insight into previous reports of a rising AKI incidence in studies based on hospital episode codes or differing AKI definitions.[10] Applying the same KDIGO-based AKI definition to data from the same region, over a 10-year span (2003–2012), the standardised AKI rates in Grampian changed little. Notably, this stability was in spite of an increasing frequency of outpatient/community testing in Grampian (whereas the frequency of hospital inpatient testing changed little over the same period). In addition, our analysis showed similar (although reduced) AKI rates across the regions when only hospital blood samples were analysed, or when the AKI definition was limited on only blood tests within the past week. Our analysis also showed a pattern of AKI phenotypes that was consistent with case-mix differences between regions. Swansea, which had the highest all-cause population mortality, also had the highest proportion of AKI phenotypes for severity, AKI progression and non-recovery.

Between-population variation in the prevalence of kidney disease has previously been described for CKD in Ireland,[37] Germany[38] and Taiwan,[39] as have variation *between* European countries.[40] In our analysis, we have now shown that much of the regional variation in AKI between UK regions can be eliminated by harmonising methods, definitions and correcting for age and sex differences. The stability we report in the AKI incidence over multiple time points in a 10-year period is contrary to previous studies from the UK and North America.[10] Given the precautions that we took to minimise heterogeneity, it is possible that some differences reported in previous studies represent a methodological artefact (eg, data capture or case-mix). Consistent with our findings, a recent study of hospital-based AKI among people admitted to the Mayo Clinic also found no significant change in AKI rates between 2006 and 2014 using a consistent creatinine change AKI definition across each year and stratifying by age and sex.[41] Furthermore, in our analysis, the pattern of differences in crude AKI rates was robust to modifications of KDIGO criteria using shorter look-back periods.

Our study has caveats common to observational studies, which we have highlighted in a conceptual model that explains the reasons for observed variation in AKI rates (figure 1). In particular, even though we used data from three regions with the same social healthcare system (the UK National Health Service), we encountered incomplete population data capture that led to the exclusion of a fourth region from the study. As we note in figure 1, differences in population capture arising out of incomplete data extraction are not necessarily visible to researchers analysing anonymised large datasets. This serves as a critical caution for researchers and policy-makers to avoid making simplistic assumptions that data from different regions are necessarily comparable when they are derived from different sources. We note that while we have used data from GP registers to provide contextual information on the populations, these data need to be interpreted carefully as they also reflect recording practices in primary care rather than solely disease burden. We would also like to remind readers that while we have applied AKI criteria consistently with the same code in each region, where sparse data exist, there still may have been bidirectional misclassification between AKI and CKD. Similarly, where AKI has occurred in the context of critical illness, falsely low creatinine values from loss of muscle mass may imply a renal recovery that has not occurred. This is a challenge for all observational studies using routine blood test data. Nevertheless, a strength of our analysis is that we have used the same pragmatic approach to this challenge across each of the populations and time periods in the study. Finally, we note that only data from three UK regions were available for inclusion in our study. This is insufficient to describe variation for the whole of the UK and other countries. This article represents a first step towards more harmonised comparisons of AKI across populations. We have shared our code with this article (https://github.com/RenalHDRUK) and now invite researchers working with population datasets in other regions to add to our experience.

In conclusion, our analysis shows the need for a robust methodological approach and recognition of case-mix differences when evaluating between-centre and temporal trends in AKI. The sharing of code is key to this approach, and we have made our code from this article available for researchers to use. Using this approach, we show strikingly similar rates of AKI across different populations from England, Scotland and Wales over a 10-year period. A consistently high burden of AKI is apparent with an estimated 1.5% of the UK population experiencing AKI each year, rising to more than 5% per year in the elderly. Current quality initiatives should adopt these methods or similar methods when evaluating the impact of changes in practice on the burden of AKI.

**Author affiliations**
[1]Institute of Applied Health Sciences, University of Aberdeen, Aberdeen, UK
[2]Farr Institute of Health Informatics Research, UK

[3]Centre for Health Informatics, Division of Informatics, Imaging and Data Sciences, Faculty of Biology, Medicine and Health, Manchester Academic Health Science Centre, The University of Manchester, Manchester, UK

[4]NIHR CLAHRC Wessex Data Science Hub, Faculty of Health Sciences, University of Southampton, Southampton, UK

[5]University of Swansea, Swansea, UK

[6]London School of Hygiene and Tropical Medicine, London, UK

[7]Academic Unit of Primary Care and Population Sciences, Faculty of Medicine, University of Southampton, Southampton, UK

[8]Usher Institute of Population Health Sciences and Informatics, University of Edinburgh, Edinburgh, UK

[9]British Heart Foundation Centre for Cardiovascular Science, University of Edinburgh, Edinburgh, UK

**Acknowledgements** We would like to acknowledge all the data providers who make anonymised data available for research. This study makes use of anonymised data held in the Secure Anonymised Information Linkage (SAIL) system, which is part of the national e-health records research infrastructure for Wales. We thank the Salford Integrated Record (SIR) board for providing us with the 2014 release of the SIR used in this study.

**Contributors** CB, SF, SS and SNvdV conceived the study. AM, GID, HAR, HOH, MJJ, SS and TMS contributed to collection of the data. AM, CB, HOH, HAR, JAC, NP, PF, PJR, SS, SNvdV, TMS and SF contributed to analysis of the data. AM, CB, DN, EM-H, GID, HOH, HAR, JAC, MJJ, NH, NP, PF, PJR, RAL, SS, SNvdV, TMS and SF contributed to interpretation of the data. SS and SF drafted the manuscript with input from AM, CB, DN, EM-H, GID, HOH, HAR, JAC, MJJ, NH, NP, PF, PJR, RAL, SNvdV and TMS. All authors approved the final version.

**Funding** This work was funded by a grant from the UK's Farr Institute for Health Informatics Research (UKHIRN/XFarrRP001). The Farr Institute is supported by a 10-funder consortium: Arthritis Research UK, the British Heart Foundation, Cancer Research UK, the Economic and Social Research Council, the Engineering and Physical Sciences Research Council, the Medical Research Council, the National Institute of Health Research, the National Institute for Social Care and Health Research (Welsh Assembly Government), the Chief Scientist Office (Scottish Government Health Directorates) and the Wellcome (MRC grant nos. CIPHER MR/K006525/1, HeRC MR/K006665/1, London MR/ K006584/1, Scotland MR/ K007017/1). We also acknowledge the data management support of Grampian Data Safe Haven (DaSH) and the associated financial support of NHS Research Scotland, through NHS Grampian investment in the Grampian DaSH. Work on this project was also part funded by Health Care and Research Wales, and by the National Institute for Health Research (NIHR) Collaboration for Leadership in Applied Health Research and Care Wessex at Southampton NHS Hospitals Foundation Trust. The views expressed are those of the authors and not necessarily those of the NHS, the NIHR or the Department of Health and Social Care. SS was supported by a research training fellowship from the Wellcome Trust to study the outcomes of acute kidney injury (WT102729/Z/13/Z).

**Disclaimer** The funders of this study had no role in study design; collection, analysis and interpretation of data; writing the report; or the decision to submit the report for publication.

**Competing interests** All authors have completed the ICMJE uniform disclosure form at www.icmje.org/coi_disclosure.pdf and declare: DN reports grants from Informatica for analyses of the National CKD Audit, which was tendered by HQIP (funding from NHS Wales and NHS England), outside the submitted work. SS is supported by a research training fellowship from the Wellcome Trust to study the outcomes of acute kidney injury (WT102729/Z/13/Z). No other support from any organisation for the submitted work; no other financial relationships with any organisations that might have an interest in the submitted work in the previous three years; no other relationships or activities that could appear to have influenced the submitted work.

**Patient consent** Not required.

**Ethics approval** Permission for the use of Grampian biochemistry data using routine biochemistry to identify AKI was provided by University of Aberdeen Sponsor, NHS Grampian Caldicott, respective data custodians, NHS Privacy Advisory Committee (ref PAC 33/14), NHS Research and Development Office (project no. 2014RM003) and National Research Ethics Service (reference 14/ NW/1371). Permission to analyse SIR data was granted to North West eHealth via the SIR approval board in 2012, which incorporates the appropriate information governance. Further ethical approval was not required due to the anonymised nature of the data. We thank the SIR board for providing us with the 2014 release

of the SIR used in this study. Permissions for using the SAIL databank were gained through application for the SAIL 0505 project, looking at acute kidney injury in Wales. This was reviewed by the Information Governance Review Panel (IGRP), which contains members of the British Medical Association (BMA), National Research Ethics Service (NRES), public health Wales, NHS Wales Informatics Service (NWIS) and consumer panel.

**Provenance and peer review** Not commissioned; externally peer reviewed.

**Data sharing statement** No additional data available. Analysis code is freely available online (https://github.com/RenalHDRUK).

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
