## [Reviewer comments · BMJ Open]

ARTICLE DETAILS

TITLE (PROVISIONAL)	Acute kidney injury in the UK: A replication cohort study of the variation across three regional populations
AUTHORS	Sawhney, Simon; Robinson, Heather; van der Veer, Sabine N.; Hounkpatin, Hilda; Scale, Timothy; Chess, James; Peek, N; Marks, Angharad; Davies, Gareth; Fraccaro, Paolo; Johnson, Matthew; Lyons, Ronan; Nitsch, Dorothea; Roderick, Paul; Halbesma, Nynke; Miller-Hodges, Eve; Black, Corrinna; Fraser, Simon

VERSION 1 – REVIEW

REVIEWER	Akira Kuriyama Kurashiki Central Hospital
REVIEW RETURNED	16-Nov-2017

GENERAL COMMENTS	This study by Sawhney aimed to examine the prevalence of AKI using a population-based database. To my knowledge, this study was the largest ever to date for this purpose. The authors tried to show a standardized prevalence of AKI. Given that this study included an extremely large sample size, it will be an important reference. Here are some comments that might improve the current manuscript. 1. Around 60 to 70% of participants did not have blood tests in the index year. This means that there were potentially many patients who were not so severe that no AKI developed in these participants. This might have underestimated the prevalence of AKI in the database used.2. The same participants also served as missing data. This might be reflected in the limitation if the authors agree with my idea.3. Studies of AKI presume that patients with missing data have eGFR= 75 based on their MDRD. I do not think that it is not always appropriate. Did the authors consider using this method in their study? The authors do not need to detail the reason in the text, but detailing this might help other reviewers understand this study.
--

REVIEWER	Norbert Lameire University hospital Gent Belgium
REVIEW RETURNED	27-Nov-2017

GENERAL COMMENTS	This is an interesting study that highlights an important topic, namely the heterogeneity in reporting AKI incidence. Authors provide age and sex adjusted AKI incidence over different time periods and across different geographical regions in the UK and provide insight in the factors that can influence AKI reporting.
---

	I have some comments that are mainly related to the algorithm used.  - Supplemental figure 2 provides a theoretical example on how the algorithm works. However the number of AKI episodes (and thus the possibility to distinguish between persistent AKI versus repetitive AKI episodes) will be dependent on the number of creatinine measurements that have been made. E.g. a patient who has two increased creatinine values (according to baseline) with an interval of three months, will be classified as having AKI that did not recover whereas we could also be dealing with two different AKI episodes with recovery in between (but not observed because creatinine has not been measured in between- thus how can you know when an new episode starts?). This highlights the problem of observational studies (mentioned by the authors in the limitations of their study) and the fact that there is a reason why patients have or don't have their creatinine measured and these reasons might differ between hospitals leading to erroneous classifications/interpretations. In the same line, if one use the first creatinine value of the index year as the baseline (no historical baseline values available), then how can one know the difference between AKI and CKD in case of an increased creatinine value? -Retrieving a baseline creatinine value by using algorithms is prone to errors. Imagine a patient hospitalized at ICU for a month with no historical baseline values available. Using the lowest value during admission as the baseline value will lead to erroneous estimation of that baseline value. Patients hospitalized at ICU will have "falsely" low values due to loss of muscle mass and thus will afterwards easily but wrongly be diagnosed as having AKI when they start regaining muscle mass and eating better and thereby increase their creatinine levels. If more ICU values are used in one cohort vs another, this might have been introduced bias. Authors should elaborate this in the limitations of the study. It is indeed not clear from this study how many ICU patients vs non ICU patient creatinine values were analyzed. -Considering the reporting of comorbidities: how were these comorbidities defined? Did you merely rely on whether or not the GP mentioned a certain comorbidity in the medical file? How reliable are incentivized GP registers? Reporting behavior might have changed across time, across regions and across individual physicians within the same time period/region. Other important variables such as 'smoking' should also be mentioned and could have influenced AKI incidence. -I find it difficult to draw any definitive conclusions about whether or not there is a difference in AKI incidence without any information on the potential difference in socio economic situation between these regions (a factor also mentioned by the authors in figure 1). Is any of these three regions considered more deprived than the others? If so, patients in that region might have avoided going to their GP or having their blood drawn because of financial reasons. This will introduce bias when estimating AKI incidence (unnoticed AKI) and when documenting comorbidities (which are also more likely to be more prevalent in the more deprived regions together with other unhealthy behavior associated with AKI such as smoking). - It is difficult to compare AKI incidence if both primary care and secondary care samples are included, with possible difference in primary vs secondary samples across time and regions (see remark above on distribution of ICU vs non ICU creatinine values). - the author s use the KDIGO (creatinine only) definition of AKI; it would have been of interest to know how many AKI (limited to 7 days) and/or AKD cases (between 7 days and 3 months) were observed.
--	---

	- although the main finding of the authors is that indeed the incidence rate of AKI, when corrected for age and sex) is not increasing over the years (at least in these 3 regions of the UK) the incidence of absolute number of AKI in elderly may be increasing in view of the changes in demography of the population. Looking at a cost perspective the cost of AKI in elderly may be different (and higher) than the rest of the population and it may still be “wise” of the public health programs to take this increased absolute number of cases of elderly AKI into account in the programming of health care. -Why did authors choose these particular NHS programs and why these particular time periods? -Is the algorithm reliable to distinguish between AKI and acute on chronic?
--	---

REVIEWER	Mark Devonald Nottingham University Hospitals NHS Trust United Kingdom
REVIEW RETURNED	11-Dec-2017

GENERAL COMMENTS	Dr Sawhney and colleagues have addressed an important problem in AKI research, namely methodological differences in studies describing incidence and outcome of AKI. Use of algorithms based on internationally recognised criteria for detection and staging of AKI, such as the KDIGO system, have facilitated comparison of studies from different regions or countries. Nevertheless, even when using the same basic AKI classification system, different investigators have used slightly different algorithms and I agree with Dr Sawhney and colleagues that investigators in this field frustratingly often do not provide adequate detail of their algorithms and methods. In my opinion this study is well designed, provides useful data and is presented clearly. I would like the authors to clarify a couple of points:  1. The study includes both inpatients and outpatients. With respect to AKI detection, these are likely to be quite different populations e.g. their frequency of serum creatinine monitoring is very different. Would the authors give their view on this, explain their preference for a population based study (rather than looking at hospitalised patients) and comment on the validity of analysing inpatients and outpatients together? Why was it not possible to distinguish outpatient and inpatient biochemistry data from Salford? 2. To detect AKI in this study, a minimum of 2 serum creatinine results within 365 days was required. Are the authors able to estimate the incidence of AKI in patients with no serum creatinine in the past 365 days i.e. patients who presented with a high serum creatinine which subsequently improved? Is this subset of patients with AKI sufficiently large to require incorporation in the study?
--

REVIEWER	Nithin Karakala University of Arkansas for Medical sciences
REVIEW RETURNED	27-Dec-2017

GENERAL COMMENTS	The topic is extremely important and will be helpful in future research studies and policy guidelines. To my knowledge it is the only study that looked at temporal trends, compared across three different geographic locations and AKI defined with different criteria. The manuscript is very well written to close attention to detail, it is a very complex paper but the authors have done a great job making it easy to follow their line of thoughts.
--

	The term of AKI phenotype is confusing as this is usually used to denote types of AKI (ischemic ATN, cardiorenal, hepatorenal, toxic ATN, etc). Please consider changing it to AKI predictors and outcomes. AKI phenotypes as an outcome could include the data in the primary results.
--	---

REVIEWER	Martin Siegel Technische Universität Berlin
REVIEW RETURNED	05-Feb-2018

GENERAL COMMENTS	Statistical review The paper addresses the incidence of acute kidney injury in the UK. I will focus on the statistics in what follows. The paper seems to be a purely descriptive paper with no inductive intentions. While not testing of differences between hospitals or years may be adequate, I miss some kind of measure of dispersion to assess the accuracy of the results. The pure use of mean values does not allow one to assess the degree of uncertainty in the results (I talk about confidence intervals around the point estimates, not about hypothesis testing and p-values). The amount of numbers presented in the tables may overstrain the reader. My feeling (although I am not a kidney specialist) is that a large part of the numbers in the tables is not really discussed in the paper. I don't exactly feel guided through the flood of numbers in the results tables and believe that this could be condensed to a much overseeable amount. I can hardly see the purpose in comparing the different (and apparently somewhat arbitrarily chosen) hospitals across years. Comparing point estimates, however, makes no sense without knowing whether or to what extent chance may play a role in fluctuations across years and towns. Finally, the conclusion that diagnoses their recording should be harmonised to allow comparisons across samples appears somewhat trivial to me.
---

VERSION 1 – AUTHOR RESPONSE

Reviewer Comment	Response
Editor Comments to Author: - Please include the study design in the title	Thank you for this suggestion. We have changed to a new title, which better reflects the study design and purpose. “Acute kidney injury in the UK: A replication cohort study of the variation across three regional populations”

	(formerly: Acute kidney injury in the UK: temporal and geographical variation in three regions)
- Please complete and include a STROBE checklist, ensuring that all points are included and state the page numbers where each item can be found.	A STROBE checklist is included and has been updated for this revision. Pages refer to the version with untracked changes.
Reviewer 1: This study by Sawhney aimed to examine the prevalence of AKI using a population-based database. To my knowledge, this study was the largest ever to date for this purpose. The authors tried to show a standardized prevalence of AKI. Given that this study included an extremely large sample size, it will be an important reference.	Thank you for appreciating the strengths and purpose of our study exploring population rates of AKI.
Here are some comments that might improve the current manuscript. 1. Around 60 to 70% of participants did not have blood tests in the index year. This means that there were potentially many patients who were not so severe that no AKI developed in these participants. This might have underestimated the prevalence of AKI in the database used.	All clinical research into kidney diseases suffers from the fact that AKI is widespread and frequently occurs unexpectedly. People do not receive blood tests in a protocolised fashion and even if they did AKI could still occur during the gaps between tests. We agree that there will be people in each population who would have been found to have AKI had we tested them. Moreover, there will even have been people with tests, who did not appear to have AKI, but would have “had AKI” if blood tests had been done on different days. All researchers need to “solve” this problem. Here we have taken a pragmatic solution from one research team, replicated it across other areas, and presented it transparently with shared code in the hope that (1) we draw attention to the barriers to doing reproducible science and (2) we can encourage other researchers follow likewise. To recognise this challenge we have added additional explanatory text to the methods and discussion sections.

	“A challenge of AKI clinical research is the operationalisation of precise international AKI criteria in “real-life” data where people do not receive blood tests in a protocolised fashion. Blood tests may not have been done at the necessary times to directly observe an acute rise in creatinine from a previous baseline, and assumptions based on available data are required. We identified differences in assumptions for determining AKI as an important potential methodological reason for observed variation in AKI rates (figure 1) and therefore used the exact same definition and analysis code in each region.” (methods section) “AKI can only be identified when sufficient blood tests have been performed to detect a change.” (methods section) “We would also like to remind readers that while we have applied AKI criteria consistently with the same code in each region, where sparse data exist there still may have been bidirectional misclassification between AKI and CKD. This is a challenge for all observational studies using routine blood test data. Nevertheless a strength of our analysis is that we have used the same pragmatic approach to this challenge across each of the populations and time periods in the study.” (discussion section)
2. The same participants also served as missing data. This might be reflected in the limitation if the authors agree with my idea.	As discussed above, the focus of this study is minimising heterogeneity in the approach to analysing real-life clinical data. We would suggest that it is not so much that data is “missing” (it is not reasonable to expect blood tests to be available for every patient every day), rather that the data that is present is there for an informative reason, such as an illness episode or health check (Goldstein Am J Epidemiol 2016). As whole population biochemistry capture has been performed, all the blood tests that were done have been included.

3. Studies of AKI presume that patients with missing data have eGFR= 75 based on their MDRD. I do not think that it is not always appropriate. Did the authors consider using this method in their study? The authors do not need to detail the reason in the text, but detailing this might help other reviewers understand this study.	Thank you. This question relates to the best approach to deal with assumptions of chronicity when data are sparse. We believe we have addressed the intent behind this question in our answers above. Specifically, existing AKI literature suggests that back calculation of MDRD-75 is not appropriate and may misclassify CKD as AKI (Siew CJASN 2013).
Reviewer 2: This is an interesting study that highlights an important topic, namely the heterogeneity in reporting AKI incidence. Authors provide age and sex adjusted AKI incidence over different time periods and across different geographical regions in the UK and provide insight in the factors that can influence AKI reporting.	Thank you for appreciating our main message.
I have some comments that are mainly related to the algorithm used.  - Supplemental figure 2 provides a theoretical example on how the algorithm works. However the number of AKI episodes (and thus the possibility to distinguish between persistent AKI versus repetitive AKI episodes) will be dependent on the number of creatinine measurements that have been made. E.g. a patient who has two increased creatinine values (according to baseline) with an interval of three months, will be classified as having AKI that did not recover whereas we could also be dealing with two different AKI episodes with recovery in between (but not observed because creatinine has not been measured in between- thus how can you know when a new episode starts?). This highlights the problem of observational studies (mentioned by the authors in the limitations of their study) and the fact that there is a reason why patients have or don't have their creatinine measured and these reasons might differ between hospitals leading to erroneous classifications/interpretations. 	This is a very eloquent description of the challenge we all face as renal epidemiologists trying to make sense of routine data to study AKI. A study using routine health data cannot capture the complete clinical context, and is at the mercy of when tests are done. Alternatively, a prospectively recruited study carries selection biases and still relies on AKI only occurring at times that it can be picked up by protocolised blood tests. Thus, we are all left with trying make the most of the data we have in real-life. The development of the definition we used here has been described elsewhere (Sawhney et al 2017 AJKD, because the focus of this paper is on replicating the code itself. With respect to the hypothetical example of an AKI episode based on one single test, with a second single high test 3 months later (which would be poor practice). No algorithm could distinguish non-recovered AKI from a second AKI episode with such sparse data, and in clinical practice a clinician would also have to deal with this uncertainty. Nevertheless, if our research is to be reproducible we need to make a pragmatic and reasonable decision. Whereas we have discussed this in our previous work, in this paper we focus on replication.

	We agree that more work is needed in this area to understand how nephrologists interpret sparse data. This would enable us to create an improved algorithm in the future that is more closely aligned to clinical practice. Nevertheless, a first step to reproducibility has to include us all working from the same recipe. In addition to the responses to reviewer 1, we have added a limitation that acknowledges that where sparse data exist there will be bidirectional misclassification between AKI and CKD. (discussion section) “We would also like to remind readers that while we have applied AKI criteria consistently with the same code in each region, where sparse data exist there still may have been bidirectional misclassification between AKI and CKD.”
In the same line, if one use the first creatinine value of the index year as the baseline (no historical baseline values available), then how can one know the difference between AKI and CKD in case of an increased creatinine value? -Retrieving a baseline creatinine value by using algorithms is prone to errors. Imagine a patient hospitalized at ICU for a month with no historical baseline values available. Using the lowest value during admission as the baseline value will lead to erroneous estimation of that baseline value. Patients hospitalized at ICU will have “falsely” low values due to loss of muscle mass and thus will afterwards easily but wrongly be diagnosed as having AKI when they start regaining muscle mass and eating better and thereby increase their creatinine levels. If more ICU values are used in one cohort vs another, this might have been introduced bias. Authors should elaborate this in the limitations of the study. It is indeed not clear from this study how many ICU patients vs non ICU patient creatinine values were analyzed.	Thank you for this question. This is why the algorithm incorporates a longer look back period for situations where baseline is being estimated and not observed. Work of Siew (CJASN 2012) and Sawhney (NDT 2015) suggests this is a reasonable approach (based on a range of sensitivity analyses), but as discussed above, not perfect. Indeed even without estimations of trend the limitations of creatinine as a static marker of kidney function are well described. Specifically relating to ITU AKI – this represented a minority of the people with AKI in our development cohort (Grampian) (10%), however not all regions were able to make this distinction. As above, this is not a unique limitation of our study, but of all clinical studies of AKI. As suggested, we have acknowledged the challenges of observational data in our discussion as outlined above. We have also added an additional line to draw attention to the issue of muscle mass in critical illness.

	“Similarly, where AKI has occurred in the context of critical illness, falsely low creatinine values from loss of muscle mass may imply a renal recovery that has not occurred.” (discussion)
-Considering the reporting of comorbidities: how were these comorbidities defined? Did you merely rely on whether or not the GP mentioned a certain comorbidity in the medical file? How reliable are incentivized GP registers? Reporting behavior might have changed across time, across regions and across individual physicians within the same time period/region. Other important variables such as ‘smoking’ should also be mentioned and could have influenced AKI incidence. -I find it difficult to draw any definitive conclusions about whether or not there is a difference in AKI incidence without any information on the potential difference in socio economic situation between these regions (a factor also mentioned by the authors in figure 1). Is any of these three regions considered more deprived than the others? If so, patients in that region might have avoided going to their GP or having their blood drawn because of financial reasons. This will introduce bias when estimating AKI incidence (unnoticed AKI) and when documenting comorbidities (which are also more likely to be more prevalent in the more deprived regions together with other unhealthy behavior associated with AKI such as smoking).	For this study we have focused purely on trying to develop biochemistry datasets in different regions that are closely aligned by using harmonised methodology. As a next step, this will open up opportunities to robustly answer a range of challenging research questions once we have combined these biochemistry datasets with additional sources of data. Table 1 provides regional QOF, renal registry and mortality statistics as a context for the populations in the absence of patient level comorbidity data available to analyse. The QOF scheme rewards GPs for keeping an active register of key conditions. This data has significant limitations as it also reflects the quality of reporting and illness behaviour leading to people presenting to their GPs to be diagnosed. We agree that recording and therefore data quality will vary and this makes it hard to interpret the data. Socioeconomic data are likely to have an association with AKI rates, but are not comparably reported in QOF across all regions and time periods. Note that the NHS is free at point of use with all patients eligible for free centralised healthcare. This is acknowledged at the bottom of figure 1. To make this important point clearer for readers we have also inserted an explanatory sentence in the methods. “All regions provide public healthcare, free at the point of use.” (methods) Given the problems with using population level statistics to describe context (as in table 1), the next step would be to link the biochemistry

	datasets here with other forms of routinely collected patient level morbidity data (such as hospital episode codes). This would be a substantial new piece of work to do across all complete populations, and will require additional ethics permissions. The potential to explore these relationships is one of the motivations for us developing this collaboration. Nevertheless, it is beyond what is possible in this first study where we have tried to align our biochemistry data analysis. We have provided the following explanations in our methods and discussion sections. “Importantly, QOF data represent incentivised recording by GPs of people with a given condition (e.g. chronic kidney disease), rather than actual population prevalences. This means that small differences in prevalence on the disease registers may represent recording practice as well as actual disease prevalence, and should be interpreted with caution.” (methods) “We note that while we have used data from GP registers to provide contextual information on the populations, these data need to be interpreted carefully as they also reflect recording practices in primary care rather than solely disease burden.” (Discussion)
- It is difficult to compare AKI incidence if both primary care and secondary care samples are included, with possible difference in primary vs secondary samples across time and regions (see remark above on distribution of ICU vs non ICU creatinine values. - the author s use the KDIGO (creatinine only) definition of AKI; it would have been of interest to know how many AKI (limited to 7 days) and/or AKD cases (between 7 days and 3 months) were observed.	Table 3 provides a range of sensitivity and subgroup analyses: We have shown that the proportion of people with AKI falls when we exclude those blood tests coming from primary care. This is likely to be because baseline function from primary care (when people are well) is likely to be more representative. In addition, a substantial proportion of AKI starts in the community. We have described rates of AKI broken down into

- although the main finding of the authors is that indeed the incidence rate of AKI, when corrected for age and sex) is not increasing over the years (at least in these 3 regions of the UK) the incidence of absolute number of AKI in elderly may be increasing in view of the changes in demography of the population. Looking at a cost perspective the cost of AKI in elderly may be different (and higher) than the rest of the population and it may still be “wise” of the public health programs to take this increased absolute number of cases of elderly AKI into account in the programming of health care.	algorithm criteria in table 3 second subsection. We believe this section provides that information in great detail. It is also provided by comparing 3 algorithm interpretations of KDIGO in figure 3. In table 4 we also provide this detail by severity, baseline, recovery and recurrence. We agree these findings are worth highlighting and have added a sentence to draw attention to this for readers: “Table 3 also shows that the majority of people developing AKI could be identified using hospital tests alone, and just over half could be identified in each region using a rigid interpretation of KDIGO AKI criteria. Finally, across all populations, the proportion of people developing AKI in the index year increased substantially with increasing age and lower eGFR.” (results) “In addition, our analysis showed similar (albeit reduced) AKI rates across the regions when only hospital blood samples were analysed, or when the AKI definition was limited on only blood tests within the past week.” (discussion) We agree with this point. As we show by reporting AKI in age strata in table 3, there is a substantial increase in AKI with age. Thus, we agree that in populations where age is increasing rapidly, policymakers should consider the cost of AKI (and long term sequelae) when planning future health care capacity. We have added this point into the first paragraph of the discussion. “The consistently high proportion of people aged over 70 developing AKI was also striking (>5%), and has implications for the planning the future health care requirements of an aging population.” (discussion)
-Why did authors choose these particular NHS	

programs and why these particular time periods?	We were limited to the few regions in the UK that had population biochemistry available to this level of capture. Only Grampian was able, at this stage, to provide biochemistry of consistent quality over multiple time periods. We recognise that this it would have been helpful to have been able to include more regions and have recognised this limitation, yet considered prioritisation of data integrity to be of vital importance for the purpose of this study. “Finally, we note that only data from three UK regions were available for inclusion in our study. This is insufficient to describe variation for the whole of the UK and other countries. This article represents a first step towards more harmonised comparisons of AKI across populations. We have shared our code with this article (https://github.com/RenalHDRUK) and now invite researchers working with population datasets in other regions to add to our experience.” (discussion)
-Is the algorithm reliable to distinguish between AKI and acute on chronic?	We, and others have discussed the performance of AKI algorithms and baseline estimation elsewhere, with acceptable distinction between AKI and CKD (Sawhney NDT 2015, Siew CJASN 2012). Distinction between AKI and “acute on chronic” also depends on baseline estimation. Numerous solutions exist and we provide one which we have used systematically in each region and justified in previous work. (Sawhney NDT 2015, Sawhney NDT 2016, Sawhney AJKD 2017)
Reviewer 3:	
Dr Sawhney and colleagues have addressed an important problem in AKI research, namely methodological differences in studies describing incidence and outcome of AKI. Use of algorithms based on internationally recognised criteria for detection and staging of AKI, such as the KDIGO system, have facilitated comparison of studies from different regions or countries. Nevertheless, even when using the same basic AKI	Thank you for appreciating the motivation behind our work.

classification system, different investigators have used slightly different algorithms and I agree with Dr Sawhney and colleagues that investigators in this field frustratingly often do not provide adequate detail of their algorithms and methods. In my opinion this study is well designed, provides useful data and is presented clearly. I would like the authors to clarify a couple of points:	
1. The study includes both inpatients and outpatients. With respect to AKI detection, these are likely to be quite different populations e.g. their frequency of serum creatinine monitoring is very different. Would the authors give their view on this, explain their preference for a population based study (rather than looking at hospitalised patients) and comment on the validity of analysing inpatients and outpatients together? Why was it not possible to distinguish outpatient and inpatient biochemistry data from Salford?	We agree - this is why we included a sensitivity analysis using only hospital data, and an analysis where we rigidly applied KDIGO criteria without any estimation if there were no blood tests in 7 days (as per responses for reviewer 2). It was not possible to include Salford in the location analysis as their blood test location stamps could distinguish community tests but not outpatient tests in the same form as the other regions. In Salford, we did not have access to hospital admission and discharge dates. In principle such data exist as a separate resource, but linking them requires complex information governance procedures and ethical approval, and we are not sure would be granted. We have briefly noted this in the statistical analysis section of the methods and in a footnote to table 2. “Of note a distinction between hospital inpatient and outpatient results was not possible in Salford.” (methods) With respect to the validity of analysing all tests, only hospital tests, broad algorithm interpretations or narrow interpretations: We would contend that there is not a right answer here as all approaches have drawbacks. What we wish to draw to the attention of researchers is that they recognise the limitations of each approach and choose an approach that most closely aligns with their research question (e.g. do they want to find all possible cases or only definite cases?). This should be clearly reported for others to replicate. We also note that the algorithm code we have shared, provides options for all these interpretations, and also has capability of being modified to change look back periods if desired. A range of interpretations of

	AKI criteria are provided in table 3, with further explanation in the results section. “Table 3 also shows that the majority of people developing AKI could be identified using hospital tests alone, and just over half could be identified in each region using a rigid interpretation of KDIGO AKI criteria.” (results) As discussed in response to reviewer 2, further discussion on patterns of testing in hospitals and the community is also available in previous work (Sawhney NDT 2016).
2. To detect AKI in this study, a minimum of 2 serum creatinine results within 365 days was required. Are the authors able to estimate the incidence of AKI in patients with no serum creatinine in the past 365 days i.e. patients who presented with a high serum creatinine which subsequently improved? Is this subset of patients with AKI sufficiently large to require incorporation in the study?	Thank you for this interesting question. We did not look at that in this study. However, elsewhere we have compared algorithm AKI with AKI coding and shown that a small number people with AKI can only be identified by this retrospective approach you suggest. In previous work we showed that the addition of a retrospective diagnosis improved the sensitivity of the NHS AKI algorithm from 91% to 95% (Sawhney NDT 2015). A methodological problem here, however, is the survival bias that would be introduced by making a special case for people who survive long enough to recover to a point where it emerges from the data that AKI has occurred.
Reviewer 4: The topic is extremely important and will be helpful in future research studies and policy guidelines. > To my knowledge it is the only study that looked at temporal trends, compared across three different geographic locations and AKI defined with different criteria. > The manuscript is very well written to close attention to detail, it is a very complex paper but the authors have done a great job making it easy to follow their line of thoughts.	Thank you for recognising the importance of this work and the subject area.
> The term of AKI phenotype is confusing as this is usually used to denote types of AKI (ischemic ATN, cardiorenal, hepatorenal, toxic ATN, etc). Please consider changing it to AKI predictors and	Thank you for raising this point. We appreciate the difficulty that phenotype could also refer to

outcomes. AKI phenotypes as an outcome could include the data in the primary results.	aetiology or to subtypes of disease mechanism (endotype). Nevertheless, phenotype can refer to any observable characteristic of a condition and we use the term in this context. It would not be appropriate to label as “predictor” as severity, recovery and baseline describe the condition itself. Similarly, outcome would not be appropriate as these characteristics become quantifiable as AKI occurs.
Reviewer 5: The paper addresses the incidence of acute kidney injury in the UK. I will focus on the statistics in what follows. The paper seems to be a purely descriptive paper with no inductive intentions. While not testing of differences between hospitals or years may be adequate, I miss some kind of measure of dispersion to assess the accuracy of the results. The pure use of mean values does not allow one to assess the degree of uncertainty in the results (I talk about confidence intervals around the point estimates, not about hypothesis testing and p-values).	We provide medians and IQRs for continuous data and have provided 95% confidence intervals around crude and standardised AKI incidences. We have provided additional detail to the key results sentence below: “Standardisation by age and sex accounted for residual differences (142-151 per 10,000/year, p value = 0.257), with 95% confidence intervals overlapping in all instances.” (results) Other results in tables are proportions. Confidence intervals could be provided, but this would make the tables very large and hard to follow.
The amount of numbers presented in the tables may overstrain the reader. My feeling (although I am not a kidney specialist) is that a large part of the numbers in the tables is not really discussed in the paper. I don’t exactly feel guided through the flood of numbers in the results tables and believe that this could be condensed to a much overseeable amount.	As can be noted by this review process, much of this information is of interest to nephrologists. In response to this suggestion and comments from reviewers above, we have added additional explanation of the tables in the body of the manuscript. We believe the table information may be useful to many readers, but at the editors’ discretion, table 4 could be moved to supplementary material. As discussed in response to reviewer 2, we have also added additional detail on the results in the tables in the results section.

I can hardly see the purpose in comparing the different (and apparently somewhat arbitrarily chosen) hospitals across years. Comparing point estimates, however, makes no sense without knowing whether or to what extent chance may play a role in fluctuations across years and towns.	We agree that a limitation of our study is that we have been restricted by the availability and quality of data available in current UK regional datasets. As we compare only three regions, we agree, and acknowledge in the discussion, that it is possible for us to have found little difference in AKI when a wide variation does in fact exist. As discussed above, we have acknowledged this in the discussion section and hope to involve additional regions in this work in the future.
Finally, the conclusion that diagnoses their recording should be harmonised to allow comparisons across samples appears somewhat trivial to me.	We wish that there was greater standardisation and transparency in how AKI (a complicated condition to code) is defined in research. However, as highlighted both in the introduction and by the reviewers, lamentably this is not the case.

VERSION 2 – REVIEW

REVIEWER	Akira Kuriyama Kyoto University, Japan
REVIEW RETURNED	05-Mar-2018
GENERAL COMMENTS	The authors have adequately revised the manuscript.
REVIEWER	Norbert Lameire University Hospital Gent, Corneel Heymanslaan , Gent, Belgium
REVIEW RETURNED	06-Mar-2018
GENERAL COMMENTS	Very nice and important study; the authors have adequately responded to my remarks and the manuscript has certainly improved.
REVIEWER	Mark Devonald Nottingham University Hospitals NHS Trust, United Kingdom
REVIEW RETURNED	18-Mar-2018
GENERAL COMMENTS	The authors have addressed my points satisfactorily, thanks. I also found their responses to the other reviewers helpful.